# A Search for Cyclin-Dependent Kinase 4/6 Inhibitors by Pharmacophore-Based Virtual Screening, Molecular Docking, and Molecular Dynamic Simulations

**DOI:** 10.3390/ijms222413423

**Published:** 2021-12-14

**Authors:** Ni Made Pitri Susanti, Sophi Damayanti, Rahmana Emran Kartasasmita, Daryono Hadi Tjahjono

**Affiliations:** 1School of Pharmacy, Bandung Institute of Technology, Jalan Ganesha 10, Bandung 40132, Indonesia; dekpitsusanti@unud.ac.id (N.M.P.S.); sophi.damayanti@fa.itb.ac.id (S.D.); e.kartasasmita@gmail.com (R.E.K.); 2Study Program of Pharmacy, Faculty of Mathematics and Natural Sciences, Universitas Udayana, Jalan Bukit Jimbaran, Badung 80361, Indonesia

**Keywords:** CDK4, CDK6, cell cycle, pharmacophore, virtual screening, molecular docking, molecular dynamic

## Abstract

The G1 phase of cell cycle progression is regulated by Cyclin-Dependent Kinase 4 (CDK4) as well as Cyclin-Dependent Kinase 6 (CDK6), and the acivities of these enzymes are regulated by the catalytic subunit, cyclin D. Cell cycle control through selective pharmacological inhibition of CDK4/6 has proven to be beneficial in the treatment of estrogen receptor-positive (ER-positive) breast cancer, particularly improving the progression-free survival of patients. Thus, targeting specific inhibition on CDK4/6 is bound to increase therapeutic efficiency. This study aimed to obtain CDK4/6 inhibitors through a pharmacophore-based virtual screening of the ZINC15 purchasable compound database using the in silico method. The pharmacophore model was designed based on the FDA-approved cdk4/6 inhibitor structures, and molecular docking was performed to further screen the hit compounds obtained. A total of eight compounds were selected based on docking results and interactions with CDK4 and CDK6, using palbociclib as the reference drug. According to the results, the compounds of ZINC585292724 and ZINC585291674 were the best compounds based on free binding energy, as well as hydrogen bond stability, and, therefore, exhibit potential as starting points in the development of CDK4/6 inhibitors.

## 1. Introduction

Cyclin-dependent kinases (CDKs) serve as the cell cycle’s main activators through phosphorylation of retinoblastoma (Rb) proteins as their key substrates. The activities of CDKs are tightly regulated by their cyclin catalytic subunit, as well as endogenous CDK inhibitors (CKI) with the ability to negatively regulate these activities. Therefore, cooperation between cyclin, CDK, and CKI is essential to ensure proper cell cycle progression [1,2]. Uncontrolled cell proliferation due to cell cycle dysregulation tends to cause subsequent cancer development, where increased expression of CDK or cyclin, as well as decreased levels of endogenous CKI, are commonly observed [3,4]. Consequently, CDKs are an effective anti-cancer therapeutic target, due to their role in cell proliferation. Among women, the most common type of breast cancer is the hormone receptor-positive/human epidermal growth factor receptor 2 negative (HR^+^/HER2^−^-) breast cancer, where *CCND1* as an encoding of cyclin D1, the catalytic subunit of CDK4 and CDK6, is profoundly expressed [5,6,7]. Therefore, targeting specific inhibition on CDK4/6 has the potential to increase therapeutic efficiency. Palbociclib is the first Food and Drug Administration (FDA)-approved CDK4/6 inhibitor with the capacity to improve progression-free survival of breast cancer patients in combination with fulvestrant or letrozole [8,9,10]. Several studies have been conducted to design and develop novel CDK4/6 inhibitors, and these have led to the synthesis of numerous compounds including 7-azabenzimidazoles [11], 4-thiazol-N-(pyridine-2-yl) pyrimidin-2-amine [12], N-(pyridin-2-yl)-4-(thiazol-5-yl) pyrimidine-2-amines [13], and piperidine sulfonamide derivatives [14].

Furthermore, screening and design of new drug candidates using computational methods have been carried out on various targets in the drug discovery and development process. In silico screening plays a significant role in increasing the efficiency of drug development, and is performed using a detailed and systematic protocol to obtain a series of hit compounds from a database. The combination of structure-based and ligand-based approaches is bound to enhance the screening’s success [15,16,17]. This study designed a computational screening method to identify novel compounds as potential CDK4/6 inhibitors. The initial stage involved screening on the ZINC15 database using a previously generated and validated ligand-based pharmacophore model. Subsequently, a molecular docking simulation was performed using two docking tools against both targets. The results obtained were then analyzed using molecular dynamic simulations as the final stage. A schematic that summarizes the overall workflow described in the present work is shown in Figure 1.

## 2. Results and Discussion

### 2.1. Pharmacophore Modelling

Pharmacophore models were built using MOE2014, based on three FDA-approved CDK4/6 inhibitors: palbociclib, ribociclib, and abemaciclib [18]. A total of 150 models containing three to seven combinations of pharmacophore features were obtained (Appendix A). However, model 77 was selected as the best based on the GH (Güner–Henry), sensitivity (Se), specificity (Sp), accuracy (Acc), and Yield of active (Ya) scores obtained from two validation steps at MOE2014 [18] and webserver Pharmitt [19], using 13 active compounds and 650 decoys as internal validation and 10 active compounds and 500 decoys as independent validation. According to Table 1, all validation steps produced scores of ≥0.7 on all parameters, indicating that model 77 is able to distinguish between active molecules, as well as decoys, and was categorized as valid [20]. This pharmacophore model comprised two hydrogen bond acceptors, one hydrogen bond donor, and two aromatic groups (Figure 2). Subsequently, the model was then used to screen the ZINC15 database [21], and a total of 6069 compounds were identified, based on the mapping of pharmacophore features. These compounds were then sequentially screened, based on drug-likeness and PAINS (Pan Assay Interference Compounds) parameters, and a total of 2251 compounds were accepted for the next screening step [22].

### 2.2. Molecular Docking Screening

To obtain reliable results, the virtual screening was performed using two docking tools, MOE2014 [18] and Autodock4.2 [23]. In addition, the docking protocols were validated by redocking the palbociclib to the active site of CDK4 and CDK6. Subsequently, the docking prediction is declared successful in cases where the root-mean-square deviation (RMSD) for the best-scored conformation is <2.0 Å [24]. The redocking of palbociclib into CDK4 using MOE2014 (triangle matcher placement scoring and London dG scoring function), as well as Autodock4.2, produced RMSD values of 0.67 and 1.35 Å, respectively. Meanwhile, the redocking of palbociclib to CDK6 using MOE2014 (alpha triangle placement scoring and alpha HB scoring function), as well as Autodock4.2, produced RMSD values of 1.19 and 0.80 Å, respectively. Therefore, these protocols have adequate accuracy for reinserting palbociclib into CDK4 and CDK6 ATP-binding pockets. The best redocking scores for palbociclib-CDK4 conformation in MOE2014 and Autodock4.2 protocols were −52.05 and −10.87, respectively, while the palbociclib-CDK6 counterparts were −57.28 and −11.58, respectively.

In addition, the receiver operating characteristic (ROC) curve was analyzed to evaluate the ability of docking protocols to discriminate between inactive and active compounds using a training set as described in Section 3.1. The results showed that MOE2014 and Autodock4.2 have a steeper ROC curve, compared to the diagonal random selection curve (Figure 3), indicating both docking protocols are able to discriminate between the inactive and active compounds. This was further verified by calculating the area under the curve (AUC) of the ROC for both docking methods. A docking method is declared to have acceptable discrimination in cases where the AUC value is ≥0.7 [25]. The AUC of MOE2014 and Autodock4.2 with CDK4 as the target were 0.879 and 0.780, respectively, while the CDK6 counterparts were 0.936 and 0.894, respectively. Therefore, the two methods are suitable for application in virtual screening.

The 2251 compounds selected based on the CDK4/6 pharmacophore features, drug-likeness, and PAINS screening were subjected to further screening by a 2-stage molecular docking. The hit compounds were first docked to CDK4 and CDK6 using MOE2014. Subsequently, the top 100 compounds with more negative docking scores than palbociclib were docked again to CDK4 and CDK6 using Autodock4.2 as the second docking tool, and 8 hit compounds were obtained as potential CDK4/6 inhibitors.

### 2.3. Molecular Docking Analysis

The eight final hit compounds were ZINC585292724, ZINC585292614, ZINC585292587, ZINC585291674, ZINC585291474, ZINC257310160, ZINC257203083, and ZINC73096242. Based on the chemical structures, most of these compounds contain a scaffold of two nitrogen (N)-containing heterocyclic rings connected with amino chains (Figure 4). These heterocyclic rings and amino chains can occupy the binding site for adenine of ATP in the hinge region [26] and have been commonly used in drug design studies for CDK4/6 inhibitors [11,12,13,14].

The conformation having the best score with CDK4 and CDK6 from docking simulation results using Autodock4.2 was analyzed to determine the interaction profile. According to the docking results, all the hit compounds were successfully docked into the ATP-binding site of CDK4 and CDK6 (Appendix A). Table 2 shows that six compounds, ZINC585292724, ZINC585292587, ZINC585291674, ZINC585291474, ZINC257203083, and ZINC73096242, formed a hydrogen bond with Val96 residue of CDK4. This residue is equivalent to Val101 residue of CDK6, recognized as a hinge region key residue in the binding of CDK6 inhibitors with pyrido [2,3-d]pyrimidinone scaffolds, such as palbociclib, ribociclib, and abemaciclib [4,27].

In addition, several compounds also formed a hydrogen bond with His95, Asp99, Asn145, and Asp158. The His95 residue, equivalent to His100 at CDK6, is the third hinge residue after the gatekeeper of the ATP-binding pocket at the kinase region’s front pocket, while the Asp99 and Asn145 are located in the ATP binding cleft of CDK4. The Asp158 residue, equivalent to Asp163 at CDK6, is a part of the DFG motif of kinase domain VII, a part of the kinase activation segment [27,28]. Of the eight hit compounds, ZINC585292614 and ZINC257310160 did not form hydrogen bonds with Val96 of CDK4. However, these two compounds have hydrophobic interactions with five and four similar residues out of seven residues of the palbociclib–CDK4 complex, and this possibly influences the compounds’ affinities towards CDK4.

Based on the molecular docking with CDK6 result (Table 3), all compounds formed a hydrogen bond with Val101, except ZINC257310160. Furthermore, four compounds, ZINC585292587, ZINC585291674, ZINC585291474, and ZINC257203083, also formed a hydrogen bond with a Lys43 residue, which is part of the K/E/D/D signature motif with important structural and catalytic roles in active protein kinases [27,29]. ZINC257310160 formed a hydrogen bond with Asp163, a part of the DFG motif of kinase domain VII in the kinase activation segment. In contrast to other compounds, ZINC585292614 also formed an additional hydrogen bond with Lys147, a part of the conserved HRD sequence of kinase domain VIB included in the catalytic loop [27,28].

### 2.4. Molecular Dynamic Simulations

#### 2.4.1. Ligand Binding Stability

The protein flexibility and the effect of solvent on the binding mode of the hit compounds were further evaluated by performing 200 ns of molecular dynamic (MD) simulations. Meanwhile, the stability of the ligand–protein complex was determined by the RMSD value during the simulation. The CDK4 backbone of each complex was equilibrated at about 0.2–0.3 nm (Appendix A). The first four complexes (Appendix A) reached the equilibration faster than others at about 150–200 ns, while the next four complexes (Appendix A) reached the equilibration at about 175–200 ns, and the palbociclib–CDK4 complex reached the equilibration at about 175–200 ns. The results suggested that the interaction of palbociclib and all the hit compounds do not have different effects on CDK4 stability. Furthermore, no significant fluctuations were observed during the 200 ns simulation in the CDK4 backbone, indicating all the protein structures remained stable.

The stabilities of the CDK6 complexes were influenced by palbociclib and all compounds at a similar level (Appendix A). The RMSD of the CDK6 backbone fluctuated at the start of the simulations and attained equilibration at about 175–200 ns. However, no significant fluctuations were observed during the 200 ns simulation in the CDK6 backbone, indicating all the protein structures remained stable. Backbone RMSD of all CDK4 and CDK6 complexes in two replicas are shown in Appendix A.

According to the ligand RMSD in the CDK4 complex (Appendix A), all the hit compounds except ZINC257310160 and ZINC73096242 showed no significant fluctuation of RMSD, indicating relatively stable docking poses. Of these hit compounds, ZINC585291674 and ZINC585291474 reached the equilibration at about 175–200 ns, while ZINC585292724 and ZINC585292587 had the average RMSD value overlap to palbociclib at about 150–200 ns and during 200 ns of MD simulations, respectively.

In the CDK6 complex (Appendix A), the most stable RMSDs were found in ZINC585292724 and ZINC585291674 with an average RMSD of 0.21 nm and 0.06 nm, respectively. In contrast to the other compounds, ZINC257310160 and ZINC257203083 exhibited a sharp increase in RMSD at around 150 ns and 100 ns, respectively. Based on the visualization, the high RMSD value observed at this time is related to the ligand conformation change, and this compound stays inside the binding pocket during simulation (see Section 2.4.2). Ligand RMSDs of all CDK4 and CDK6 complexes in two replicas are shown in Appendix A.

The root mean square fluctuation (RMSF) analysis was carried out to discover the flexibility of amino acid residues on the targets during the simulation. CDK activation begins with cyclin binding, which continues with activation of the ATP-binding site. In the next step, the cyclin takes the C-lobe of the activation segment out of the catalytic site. This causes the T-loop (threonine residue) to be phosphorylated by CDK-activating kinase (CDK7-cyclin H complex) to the complete active form. The conserved DFG motif and APE sequence constitute the beginning and end of the activation segment (^158^DFG^160^-^182^APE^184^ of CDK4 and ^163^DFG^165^-^187^APE^189^ of CDK6). This activation segment includes the T-loop (T172 of CDK4 and T177 of CDK6). In protein kinase, ATP binds to the catalytic cleft between N- and C-lobes, which are divided into three regions, front pocket (FP), gate area, and back cleft. FP consists of a gly-rich loop, hinge residues (Glu93-Val96 for CDK4 and Glu99-Val101 for CDK6), an ATP-binding pocket, the extension connecting the C-lobe’s αD-helix to the hinge residues as well as amino acid residues in the catalytic loop. The gate area comprises the activation segment’s proximal portion, including the β3-strand and DFG motif. The back cleft extends to the αC-helix and αE-helix within the C-lobe, portions of the N-lobe’s β4- and β5-strands, as well as the αC-β4 back loop. The gate area and back loop forms the back pocket (BP) [4,27]. Figure 5 shows the 3D structure and ATP binding site of CDK4 and CDK6.

Figure 6 and Figure 7 show the residues of CDK4 and CDK6 in the hinge region, as well as the DFG motif in domain VII and conserved APE sequence in domain VIII, constituting the activation segment’s beginning and end, which are quite stable with low RMSF values. This is probably due to the interaction between the eight hit compounds within these areas, making the compounds have seemingly less fluctuation. This is also supported by the RMSF results shown by the CDK4 and CDK6 backbones without ligand interactions which fluctuate with significant RMSF, especially in the activation segment. Generally, each hit compound and palbociclib as a reference showed similar RMSF profiles, indicating these compounds influence each target at the same level.

Analysis of key ligand–protein distances evolutions throughout the simulations was carried out on several important residues of CDK4 and CDK6, namely the third hinge residue (Val96 for CDK4 and Val101 for CDK6), the second hinge residue (His95 for CDK4 and His100 for CDK6), K (Lys) of K/E/D/D motif (Lys35 for CDK4 and Lys43 for CDK6), D (Asp) of the DFG motif (Asp158 for CDK4 and Asp163 for CDK6), and the first residue of the ATP-binding cleft (Asp99 for CDK4 and Asp104 for CDK6) [4,27,28]. The list of ligand-residue interactions is shown in Appendix A.

On the CDK4 complex (Figure 8), palbociclib did not show significant distance fluctuations with all important residues during the simulation. The largest fluctuation occurred in the distance to the His95 residue. Distance fluctuations with residue occurred in all hit compounds. In the compounds ZINC585292724 and ZINC585292614, significant fluctuations were shown in the distance to the Asp99 residue. Distance fluctuations with Asp158 residues occurred in all hit compounds except ZINC585292724, ZINC585292614, and ZINC257203083, while significant distance fluctuations with Lys35 residue occurred in ZINC585291474, ZINC257310160, and ZINC257203083 compounds. In complexes with CDK6 (Figure 9), all compounds exhibited distance fluctuations with Asp163 residue. Significant fluctuations with His100 residue occurred in ZINC585292724 and ZINC585292587 compounds. Significant distance fluctuations occurred in all residues with ZINC257310160, ZINC257203083, and ZINC73097242 compounds. Despite fluctuations in the distance between the hit compounds and residues, all hit compounds persisted inside the ATP-binding pocket of CDK4 and CDK6 during the simulations (see Section 2.4.2).

Of all the hit compounds, ZINC585292724, ZINC585292587, ZINC585291674, ZINC585291474, and ZINC73097242 were able to maintain the distance with Val96 residue, while ZINC585292724, ZINC585292614, ZINC585292587, ZINC585291674, and ZINC585291674 had a stable distance with Val101 residue during the simulation. These two residues are key residues in the binding of CDK4/6 inhibitors. These results are in line with the results of the hydrogen bond stability analysis (see Section 2.4.2).

#### 2.4.2. Hydrogen Bond Analysis

The occupancy percentages of hydrogen bonds between electronegative donor and acceptor atom at 200 ns of MD simulations were calculated for all complexes, and the hydrogen bonding is said to possess adequate stability in cases where the occupancy is above 50% [30]. Based on the results, five compounds, ZINC585292724, ZINC585292587, ZINC585291674, ZINC585291474, and ZINC73096242, form stable hydrogen bonds in complexes with CDK4. Meanwhile, the CDK6 counterparts are ZINC585292724, ZINC585292614, ZINC585292587, ZINC585291674, and ZINC585291474 (Table 4).

The stable hydrogen bonds were formed with the key residues of both targets: the third hinge residue after the gatekeeper. For CDK4 complexes, similar to palbociclib, all five compounds showed hydrogen bonds with Val96 through the N atom of N-containing aromatic rings as donors with occupancy of 52.26%, 51.14%, 55.78%, 57.26%, and 53.45, respectively. Meanwhile, for the CDK6 complexes, all five compounds formed hydrogen bonds with the same residue as palbociclib, Val101, through the N atom of N-containing aromatic rings as donors with occupancy 67.38%, 73.29%, 80.43%, 86.31%, and 68.33%, respectively. Therefore, the results generally indicate these potential compounds tend to exhibit better binding interactions with CDK6, compared to CDK4.

Based on hydrogen bond occupancy, analysis of bond distance during the simulation was carried out on bonds with occupancy above 50%. The results are shown in Appendix A. In line with the hydrogen bond occupancy rate, the hydrogen bond distance throughout the simulation time shows an increase in stability as the occupancy increases. The hydrogen bond with the highest stability was shown by the bond between N59 of the compound ZINC585291674 with the O atom of Valine101 residue of CDK6. Figure 10 shows the comparison of the hydrogen bond distances between compounds with Val69 of CDK 4 and Val 101 of CDK6 residues during the simulation. ZINC585292614 did not form hydrogen bonds with Val96 residue of CDK4 whereas ZINC257310160 did not form hydrogen bonds with these two residues.

Figure 11 shows that all the compounds remained in the CDK4 binding pocket, including ZINC585292614, ZINC257310160, and ZINC257203083 with hydrogen bond occupancy below 50%. However, the positions and conformations of these three compounds changed at the end of the simulation. Meanwhile, the other five compounds with hydrogen bond occupancy above 50% exhibited no change in position or conformation during the 200 ns simulation. All the compounds also appear to persist in the CDK6 binding pocket (Figure 12). However, ZINC257310160, ZINC257203083, and ZINC73096242 with hydrogen bond occupancy below 50% exhibited changes in positions and conformations at the end of the simulations. Meanwhile, the other five compounds with hydrogen bond occupancy above 50% exhibited no change in position or conformation during the 200 ns simulation. This implies a low percentage of hydrogen bond occupancy is associated with poor interaction stability.

#### 2.4.3. MMPBSA Analysis

The Molecular Mechanics Poisson–Boltzmann Surface Area (MMPBSA) analysis was performed to calculate the binding free energy of the eight potential compounds during steady-state. Table 5 and Table 6 show the van der Waals, electrostatic, and enpolar energies affecting ligand binding.

In the interaction with CDK4, compounds ZINC585292724 and ZINC585291674 had the lowest free binding energies, even compared to palbociclib. This is in line with the results of the interaction at both the docking stage and MD simulation. The visualization also showed that these compounds remained in the ATP-binding pocket of CDK4 during the 200 ns simulation and showed no significant conformational change (Figure 11). Similar results were shown in the interaction with CDK6, and this is possible because CDK4 and CDK6 are functional homologs.

For the CDK6 complexes, five compounds (ZINC585292724, ZINC585292587, ZINC585291674, ZINC585291474, and ZINC257310160) obtained better free binding energy, compared to palbociclib. This indicates the compounds’ ability to bond to CDK6 with a good affinity. Based on the interaction profile, only ZINC257310160 formed no form of hydrogen bond with hinge residue Val101 of CDK6, at both the docking stage and MD simulations.

Compounds ZINC585291674 and ZINC585292724 exhibited the lowest and second-lowest binding energies, respectively, of all the compounds. According to the visualization, these compounds occupied an area of ATP-binding pocket and exhibited no change in position or conformation during the 200 ns simulation (Figure 12). Based on several parameters, for instance, binding energy, RMSD profiles, hydrogen bond stability, and binding mode to the ATP-binding pocket (Figure 13), ZINC585292724 and ZINC585291674 were selected for further development as potential CDK4 and CDK6 inhibitors.

## 3. Materials and Method

### 3.1. Ligand-Based Pharmacophore Modelling

The 3D structures of three FDA-approved inhibitors palbociclib, ribociclib, and abemaciclib were extracted from the X-ray crystallography structure of CDK6 (PDB ID: 5L2I, 5L2T, and 5L2T) downloaded from the RCSB protein data bank (PDB) (https://www.rcsb.org/) (accessed on 30 June 2021). Subsequently, pharmacophore modeling was carried out by aligning these molecules using flexible alignment in MOE2014 (Chemical Computing Group ULC, Montreal, QC, Canada) [31]. A total of 150 pharmacophore models were generated by combining three to seven pharmacophore features. Of these models, model 77 was selected for the screening stage based on the validation result of the ability to distinguish active compounds from decoys by screening a training set for internal validation and test set for independent validation. This training set comprised 13 known active compounds selected from published literature [11,12,13] including palbociclib, ribociclib, and abemaciclib and 650 decoy compounds for CDK4 and CDK6 targets. The test set comprised 10 active known active compounds from the published literature [11,12,13] excluding palbociclib, ribociclib and abemaciclib, and 500 decoys. Decoy compounds were obtained from ZINC15 (https://zinc15.docking.org/) (accessed on 8 July 2021) [21] database using Decoyfinder software (Centre Tecnològic de Nutrició i Salut (CTNS), Reus, Catalonia, Spain) [32]. Compounds are considered to be decoys if the following conditions are met: they are similar to the active molecule according to five physical descriptors: (i) the molecular weight is within 25 Da of the active ligand; (ii) they contain the same number ± 1 of rotational bonds and hydrogen bond donors, and the same number ± 2 of hydrogen bond acceptors; and (iii) the LogP value is within 1.0 of the active ligand. The Tanimoto coefficients between a potential decoy and each of the active molecules are not greater than 0.75. The Tanimoto coefficients between a potential decoy and previously selected decoys are not greater than 0.9 [33].

The previously obtained pharmacophore model was subjected to an initial screening using Pharmitt (https://pharmit.csb.pitt.edu/) (accessed on 19 July 2021) [19], to search for hits from the ZINC purchasable database comprising 13,190,317 compounds. Furthermore, the input parameters were 500 Daltons maximum molecular weight, 5.0 LogP, five hydrogen bond donors, 10 hydrogen bond acceptors, 10 rotatable bonds, and 140 polar surface area cut-off. Furthermore, the database of hit compounds was downloaded for PAINS screening [22] and further screening.

### 3.2. Molecular Docking

The hit compounds were subjected to further screening by molecular docking simulations on CDK4 and CDK6 as targets using MOE204 [18] and Autodock4.2 (The Scripps Research Institute, La Jolla, CA, USA) [23]. In addition, the crystal structures of CDK4 (PDB ID: 2W96) and CDK6 (PDB ID: 2EUF) were downloaded from the RCSB protein data bank (PDB). The 2EUF is a CDK6 structure in complex with palbociclib as a native ligand, while the 2W96 is CDK4 without a native ligand. The site-specific docking in Autodock4.2 towards known residues of the CDK4 ATP-binding pocket was performed using palbociclib as a ligand, to obtain the ligand-binding pocket of CDK4. The MD simulation of the docking results was carried out at 100 ns until a steady-state condition was obtained. Subsequently, this stable conformation was used as a model in subsequent molecular docking.

Molecular docking using MOE2014 was performed through rigid receptor docking where the CDK4 and CDK6 were set as rigid, while the ligands were set as flexible. Docking site option was set as ligand atom. Furthermore, CDK4 and CDK6 were prepared using the LigX module, while the training set and decoys were assigned a partial charge of MMFF94X. The docking protocol was then performed by simulating the molecular docking of native ligands to the target protein by combining the placement scoring and scoring function in the MOE docking protocol [18].

Proteins without water molecules and palbociclib as ligand were separated using Chimera (University of California, San Francisco, CA, USA) [34] before preparation for docking using Autodock4.2. The protein and ligand preparation were carried out with AutodockTools and involved setting rotatable bonds, merging nonpolar hydrogen bonds, as well as adding gasteiger charges. Polar hydrogen was also added to the entire protein, while the 40 × 40 × 40 Å grid box dimensions with a grid spacing of 0.375 Å was set by AutoGrid module of Autodock4.2 which covers the entire protein active site region. The protein and ligand were rewritten into PDBQT format, and the docking protocol was set with the protein set at fixed and the ligands set at flexible. Subsequently, the electrostatic map and affinity maps for all the atom types present were computed. Lamarckian Genetic Algorithm (GA 4.2) was used for the docking, and this provides the top 10 estimated free energy of binding scores for each trial.

To validate the accuracy of both docking protocols, palbociclib was redocked to the binding pocket of protein and the RMSD value was calculated. The docking performance was also validated by analyzing the AUC of ROC after screening the set of active and decoys compounds.

### 3.3. Molecular Dynamic Simulations

Molecular dynamic (MD) simulations were performed in two replicates minimum using the Amber 16 package (University of California, San Francisco, CA, USA) [35]. The preparation file was generated using antechamber and parmchk modules of Amber16. MD simulations at 200 ns were performed using Generalised Amber Force Field (GAFF) and FF14SB for the complex of CDK4 and CDK6 with all the hit compounds. Subsequently, the complexes were protonated and the total charge was zeroed by adding an appropriate counter ion (Na^+^). TIP3PBOX water model was used to solvate the molecules with the box’s edge at least 10 Å away from the solute molecules. Each complex system was also maintained at 310 K with constant pressure throughout the simulation.

Energy minimization was done for 5000 steps, and the MD simulation was carried out using the Particle Mesh Ewald Molecular Dynamics (PMEMD) of Amber16. The trajectories were collected for every 10 ps to obtain insights into the interactions at the atomistic level. Meanwhile, the Root Mean Square Deviation (RMSD) and hydrogen bond average were calculated by the CPPTRAJ module of the Amber tool [36]. Furthermore, the mmpbsa.py tool of Amber16 was used to calculate the binding energy of each complex from the MD trajectories [37]. The binding energies were also calculated based on the 400 extracted snapshots from the last 20 ns trajectory of 200 ns simulation results.

## 4. Conclusions

This study developed in silico methods employing ligand-based pharmacophore screening, molecular docking, and molecular dynamic simulations, to identify the potential compounds of CDK4 and CDK6 inhibitors from the ZINC database. The results showed eight compounds matching the mapping of pharmacophore features with docking scores similar to palbociclib as a reference drug. Based on the binding mode analysis during molecular docking, seven of these compounds exhibited similar interactions to palbociclib toward CDK4 and CDK4. Meanwhile, the analysis of binding mode during molecular dynamic simulations showed the two compounds ZINC585292724 and ZINC585291674 possessed stable hydrogen bonds with the essential residues of CDK4 and CDK6 as well as good binding free energy, indicating steady interaction to the ATP-binding pocket of CDK4 and CDK6. Therefore, ZINC585292724 and ZINC585291674 have potential as CDK4 and CDK6 inhibitors for further development. Currently, biological activity studies of these two compounds against CDK4 and CDK6 are ongoing and will be reported in a separate article.

## Figures and Tables

**Figure 1 ijms-22-13423-f001:**
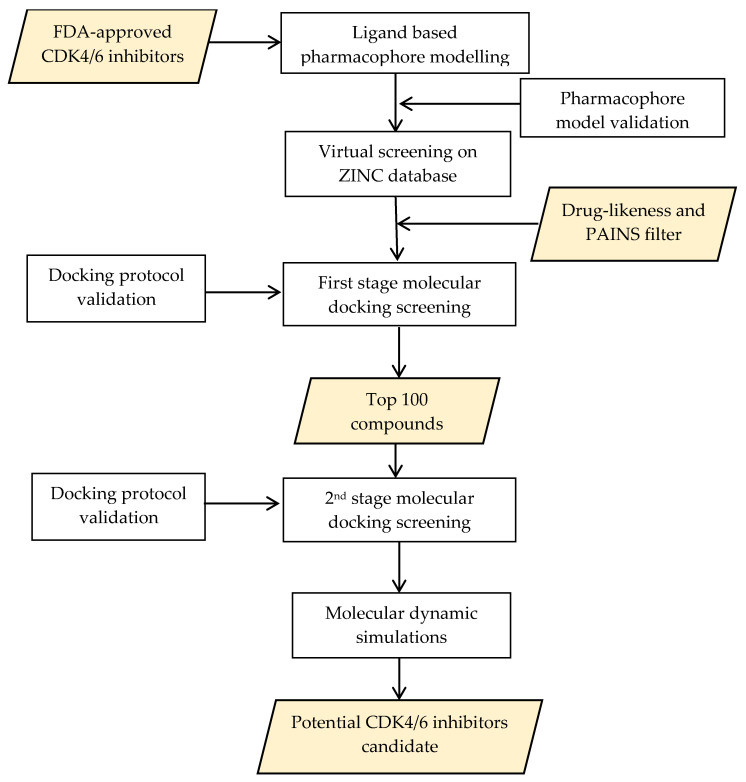
Schematic summarizing the overall workflow described in this work.

**Figure 2 ijms-22-13423-f002:**
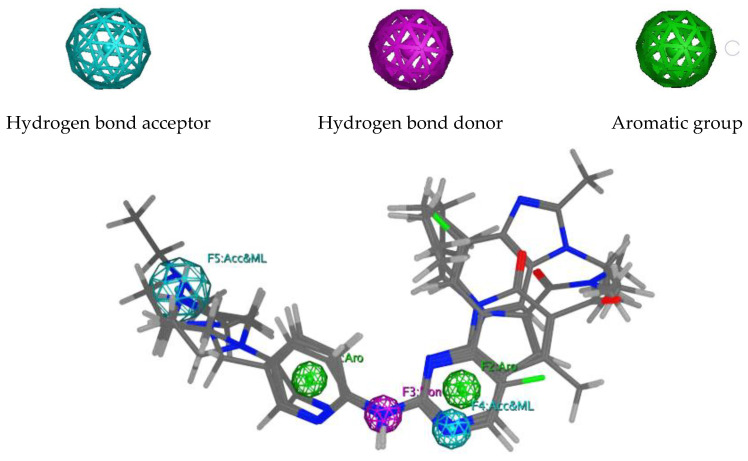
An overlay of the aligned structure of palbociclib, ribociclib, and abemaciclib on pharmacophore model 77.

**Figure 3 ijms-22-13423-f003:**
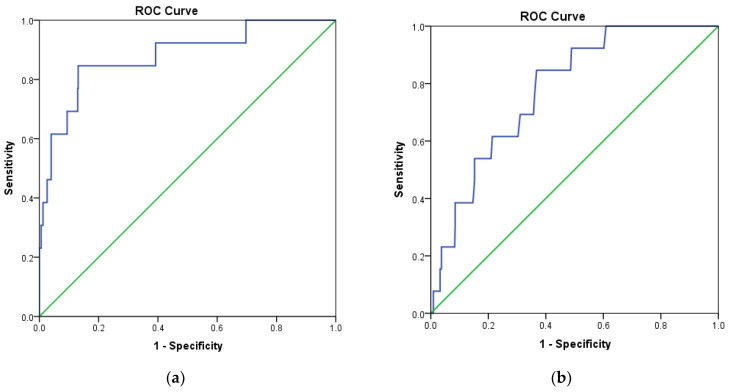
Receiver operating characteristic (ROC) curve validation of docking protocols using MOE2014 and Autodock4.2. (**a**) CDK4 with MOE2014, (**b**) CDK4 with Autodock4.2, (**c**) CDK6 with MOE2014, (**d**) CDK6 with Autodock4.2.

**Figure 4 ijms-22-13423-f004:**
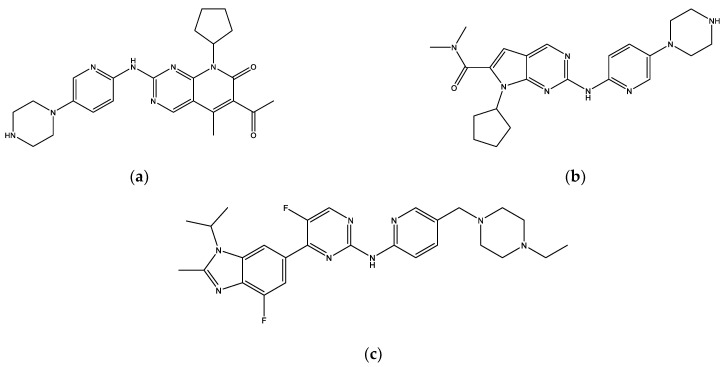
The chemical structures of FDA-approved drugs (**a**) Palbociclib, (**b**) Ribociclib, (**c**) Abemaciclib, and eight hit compounds (**d**) ZINC585292724, (**e**) ZINC585292614, (**f**) ZINC585292587, (**g**) ZINC585291674, (**h**) ZINC585291474, (**i**) ZINC257310160, (**j**) ZINC257203083, and (**k**) ZINC73096242.

**Figure 5 ijms-22-13423-f005:**
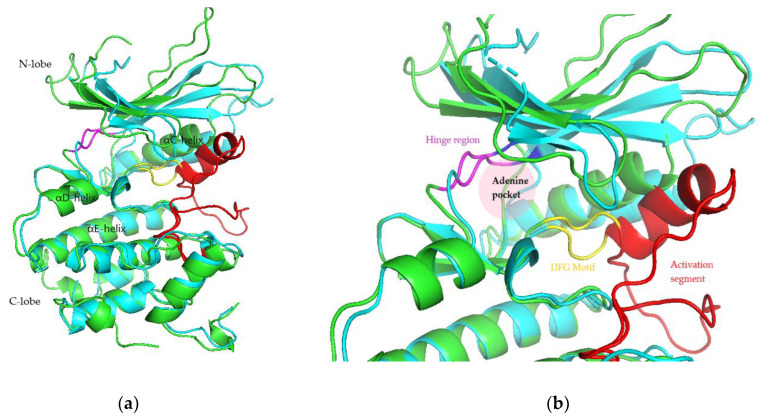
3D structure (**a**) and ATP binding site (**b**) of CDK4 (green) and CDK6 (cyan).

**Figure 6 ijms-22-13423-f006:**
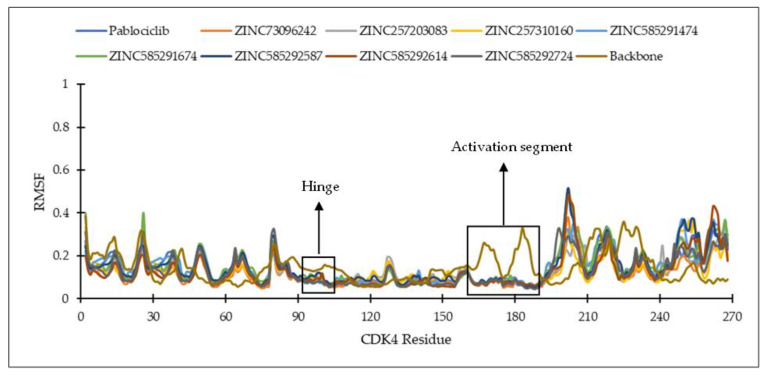
The RMSF of the amino acid residue of CDK4 during the MD simulation.

**Figure 7 ijms-22-13423-f007:**
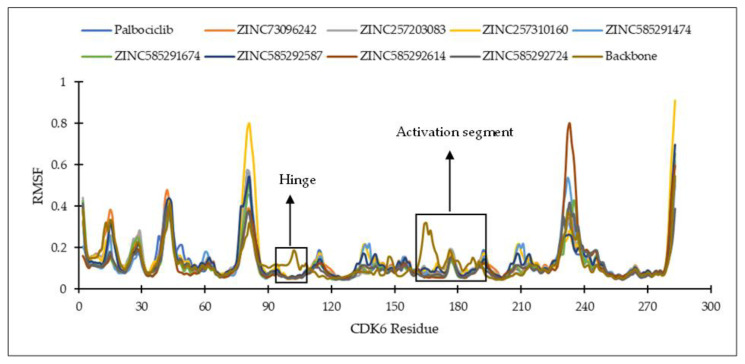
The RMSF of the amino acid residue of CDK6 during the MD simulation.

**Figure 8 ijms-22-13423-f008:**
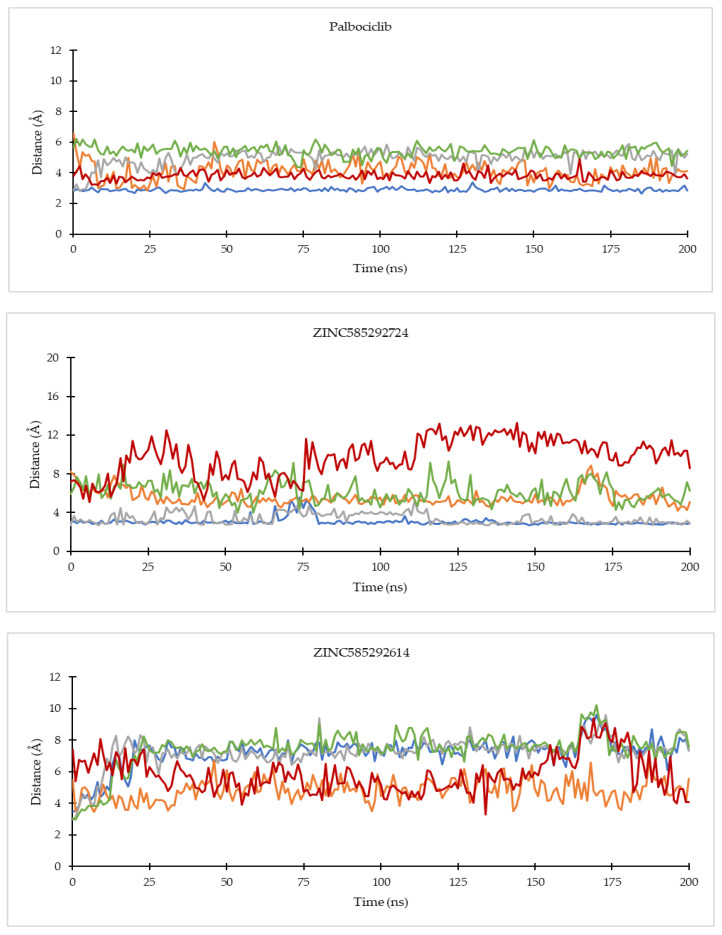
The evolution of the mean ligand–CDK4 residue distances of two replicas throughout the simulations. Color code: Blue: Val96; Orange: His95; Grey: Lys35; Green: Asp158; Red: Asp99.

**Figure 9 ijms-22-13423-f009:**
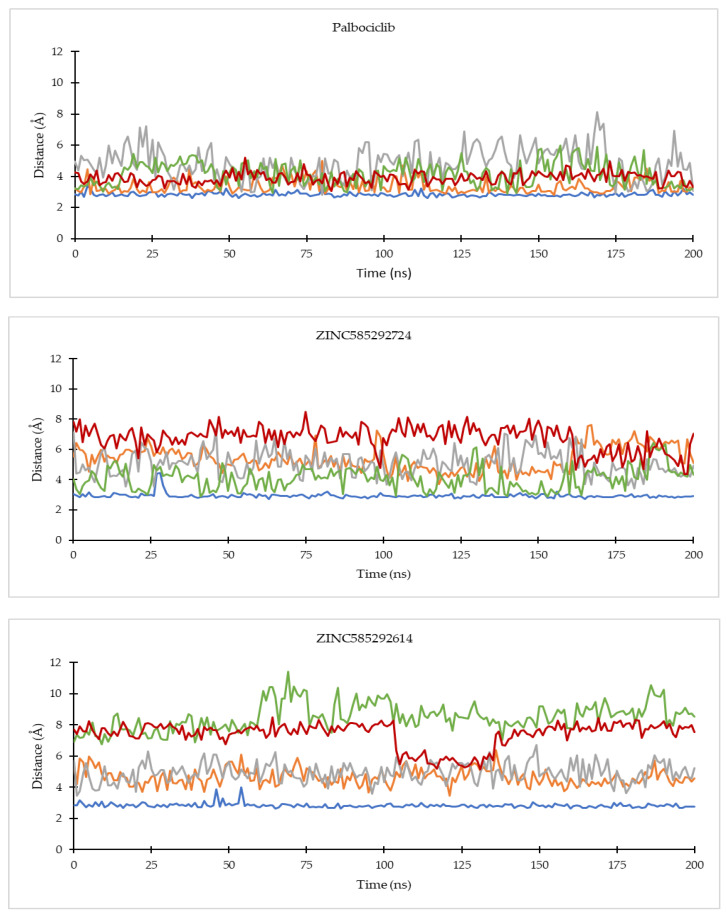
The evolution of the mean ligand–CDK6 residue distances of two replicas throughout the simulations. Color code: Blue: Val101; Orange: His100; Grey: Lys43; Green: Asp163; Red: Asp104.

**Figure 10 ijms-22-13423-f010:**
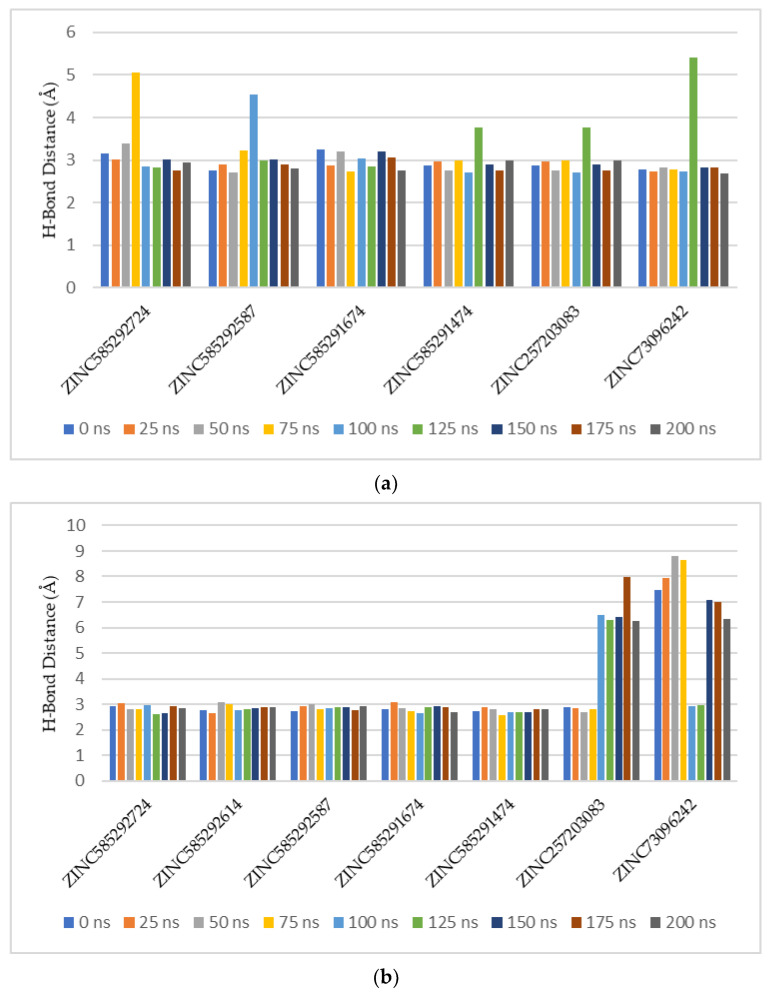
Comparison of the hydrogen bond distances between compounds with Val69 of CDK4 (**a**) and Val101 of CDK6 (**b**) residues during the simulation.

**Figure 11 ijms-22-13423-f011:**
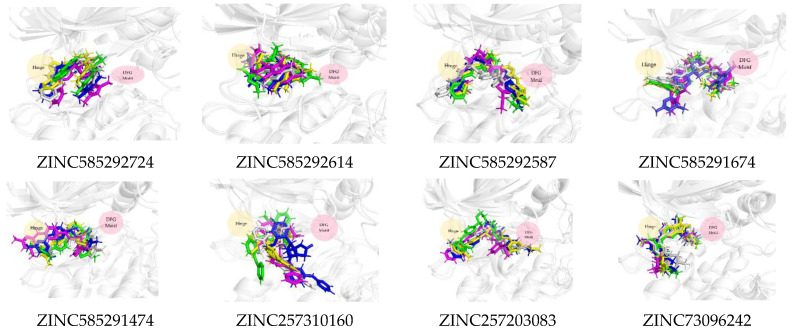
The superimposed ligand binding poses of hit compound at 0 ns (green), 100 ns (grey), and 200 ns (blue) in complexes with CDK4.

**Figure 12 ijms-22-13423-f012:**
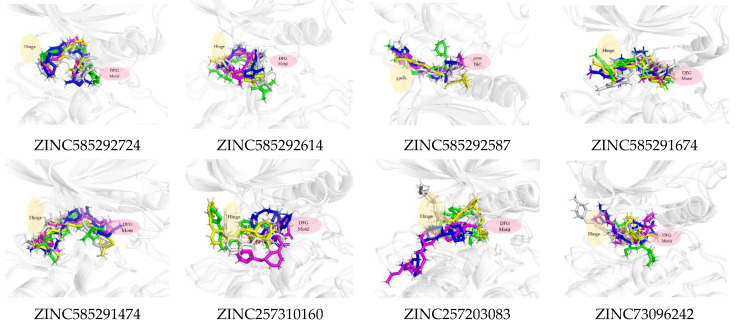
The superimposed ligand binding poses of hit compound at 0 ns (green), 100 ns (grey), and 200 ns (blue) in complexes with CDK6.

**Figure 13 ijms-22-13423-f013:**
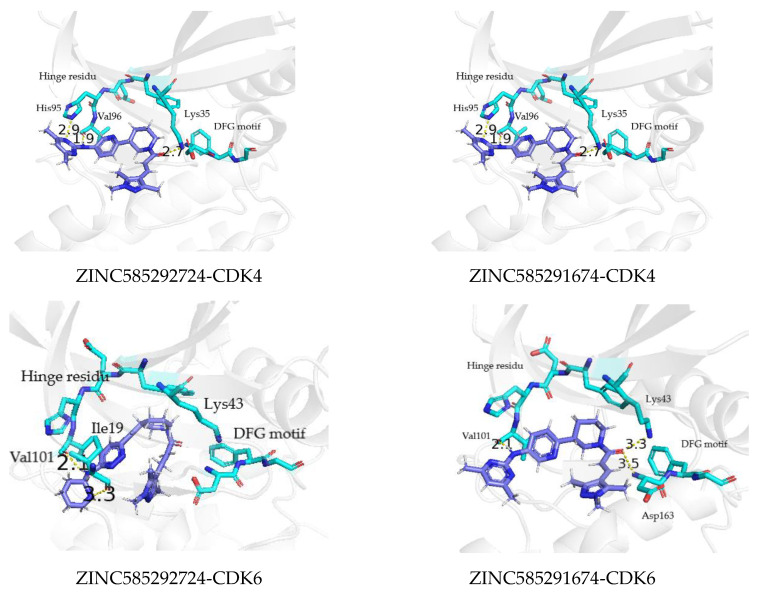
The binding mode of ZINC585292724 and ZINC585291674 to the ATP binding pocket of CDK4 and CDK6 at 200 ns.

**Table 1 ijms-22-13423-t001:** The validation results for pharmacophore model 77.

	Internal Validation					Independent Validation				
	GH	Se	Sp	Acc	Ya	GH	Se	Sp	Acc	Ya
MOE2014	0.981	0.923	1.000	0.998	1.000	0.975	0.900	1.000	0.998	1.000
Pharmitt	0.981	0.923	1.000	0.998	1.000	0.975	0.900	1.000	0.998	1.000

GH: Güner–Henry; Se: Sensitivity; Sp: Specificity, Acc: Accuracy; Ya: Yield of active.

**Table 2 ijms-22-13423-t002:** The CDK4–ligand interactions recorded during docking.

Ligands	Docking Score MOE2014	Docking Score Autoock4.2	H-Bond	Hydrophobic Interaction
Palbociclib	−51.94	−10.77	Val96 Asp99	Ile12, Val20, Ala33, Val96, Val72, Leu147, Ala157
ZINC585292724	−54.01	−11.11	Val96	Ile12, Ala33, Leu147, Ala157
ZINC585292614	−52.13	−10.84	Asp158	Ile12. Tyr17, Val20, Ala33, Val72, Phe93, His95, Leu147
ZINC585292587	−53.07	−11.03	Val96 Asn145	Tyr17, Val20, Ala33, Val72, Leu147, Ala157
ZINC585291674	−54.88	−11.29	Val96	Ile12, Val20, Ala157, Leu147
ZINC585291474	−53.03	−11.15	Val96	Ile12, Lys22, Ala33, His95, Val96, Ala157
ZINC257310160	−52.86	−10.82	Asp99	Ile12, Val20, Val96, Leu147
ZINC257203083	−52.62	−11.02	Val96 His95	Ile12, Ala16, Val20, His95, Asp97, Leu147
ZINC73096242	−53.23	−10.86	Val96 His95	Ile12, Val20, Ala33, Val72, Val96, Asp97, His95, Leu147, Ala157

**Table 3 ijms-22-13423-t003:** CDK6–ligand interactions recorded during docking.

Ligands	Docking Score MOE2014	Docking Score Autoock4.2	H-Bond	Hydrophobic Interaction
Palbociclib	−57.16	−11.50	Val101 Asp163	Val27, Ala41, Val77, Phe98, Ile19, Leu152, Ala162
ZINC585292724	−59.03	−11.62	Val101	Ile19, Val27, Ala41, Lys43, Phe98, Gln103, Leu152, Ala162
ZINC585292614	−60.28	−11.74	Val101 Lys147	Ile19, Val27, Asp102, Gln103
ZINC585292587	−59.23	−11.62	Val101 Lys43	Ile19, Val27, Ala41, Val77, Phe98, Val101, Leu152, Ala162
ZINC585291674	−62.01	−12.28	Val101 Lys43	Ile19, Val27, Ala41, Phe98, Val101, Gln103, Leu152, Ala162
ZINC585291474	−61.65	−12.18	Val101 Lys43	Val27, Ala41, Lys43, Val77, Phe98, His100, Leu152, Ala162
ZINC257310160	−59.03	−11.52	Asp163	Ile19, Val27, Ala41, Phe98, Val101, Leu152, Ala162
ZINC257203083	−59.26	−11.50	Val101 Lys43	Ile19, Val27, Tyr108, Lys111, Leu152, Val153
ZINC73096242	−61.85	−11.81	Val101	Ile19, Tyr24, Val27, Lys43, Ala41, His100, Leu152

**Table 4 ijms-22-13423-t004:** The occupancy of hydrogen bond during 200 ns simulations.

Ligands	Target			
	CDK4		CDK6	
	Donor-Acceptor	Occupancy (%)	Donor-Acceptor	Occupancy (%)
Palbociclib	(N21)-VAL96(O) (N31)-Thr102(O) Val96(N)-(N9)	71.65 26.04 13.88	(N21)-Val101(O) His100(H2)-(N23) (N31)-Thr107(O)	82.15 28.36 9.74
ZINC585292724	(N56)-Val96(O) Lys35(N)-(O61)	52.26 20.26	(N56)-Val101(O)	67.38
ZINC585292614	Asp158(N)-(N64) (N60)-Asp158(O)	2.54 2.47	(N60)-Val101(O)	73.29
ZINC585292587	Val96(N)-(O52) Lys35(N)-N51	51.14 10.69	(N46)-Val101(O)	80.43
ZINC585291674	(N59)-Val96(O) Lys35(N)-(O64)	55.78 29.39	(N59)-Val101(O) Lys43(N)-(O64)	83.61 26.59
ZINC585291474	(N51)-Val96(O) Lys35(N)-(O56)	57.26 17.46	(N51)-Val101(O)	68.33
ZINC257310160	(N61)-Glu144(O) Tyr17(OH)-(O56)	10.42 8.94	(N59)-Asp104(O) (N61)-Asp102(O)	34.39 13.66
ZINC257203083	(N58)-Val96(O) Lys35(N)-(O55) His95(N)-(N60)	33.22 24.5 22.02	(N58)-Val101(O) (N58)-Asp144(O)	35.25 17.31
ZINC73096242	(N60)-Val96(O) His95(N)-(o56) Lys35(N)-(o58) Val96(N)-(N62)	53.45 35.0 24.02 9.83	(N60)-Asp104(O) (N60)-Val101(O) (N60)-Asp163(O)	23.05 17.21 9.82

**Table 5 ijms-22-13423-t005:** The predicted binding free energy and the individual energy components (Kcal/mol) for the ligand–CDK4 complexes.

Ligands	Van der Waals Energy (∆*E*_VDW_)	Electrostatic Energy (∆*E*_elec_)	EPB (∆*E*_polar_)	Enpolar (∆*E*_non polar_)	Binding Energy (∆*G*_bind_)
Palbociclib	−55.52 ± 2.62	−30.85 ± 7.84	56.3 ± 6.26	−4.61 ± 0.12	−34.68 ± 3.46
ZINC585292724	−57.47 ± 4.24	−118.27 ± 13.53	141.94 ± 11.17	−3.89 ± 0.11	−37.69 ± 4.32
ZINC585292614	−41.67 ± 4.11	−129.2 ± 12.56	145.89 ± 11.60	−3.33 ± 0.18	−28.31 ± 4.30
ZINC585292587	−56.19 ± 2.71	−25.84 ± 5.43	51.88 ± 3.86	−4.67 ± 0.12	−34.81 ± 5.07
ZINC585291674	−55.29 ± 3.24	−27.40 ± 4.71	48.14 ± 4.38	−3.86 ± 0.07	−38.41 ± 3.84
ZINC585291474	−57.22 ± 3.15	−43.69 ± 5.69	68.55 ± 4.91	−5.17 ± 0.11	−37.53 ± 4.61
ZINC257310160	−44.57 ± 4.27	−27.85 ± 5.67	49.33 ± 5.08	−4.66 ± 0.25	−27.75 ± 4.19
ZINC257203083	−47.82 ± 4.23	−27.91 ± 8.12	49.92 ± 7.81	−4.26 ± 0.21	−30.08 ± 4.35
ZINC73096242	−50.94 ± 3.18	−24.37 ± 45.70	50.20 ± 6.73	−4.65 ± 0.19	−29.75 ± 4.54

**Table 6 ijms-22-13423-t006:** The predicted binding free energy and the individual energy components (Kcal/mol) for the ligand−CDK6 complexes.

Ligands	Van der Waals Energy (∆*E*_VDW_)	Electrostatic Energy (∆*E*_elec_)	EPB (∆*E*_polar_)	Enpolar (∆*E*_non polar_)	Binding Energy (∆*G*_bind_)
Palbociclib	−55.13 ± 3.21	−29.12 ± 4.87	56.98 ± 4.30	−4.81 ± 0.12	−32.08 ± 4.90
ZINC585292724	−51.50 ± 4.18	−99.09 ± 11.01	118.10 ± 10.01	−4.49 ± 0.14	−36.99 ± 5.31
ZINC585292614	−54.04 ± 3.13	−28.46 ± 3.96	57.23 ± 4.09	−4.59 ± 0.11	−29.87 ± 4.56
ZINC585292587	−54.58 ± 2.75	−23.03 ± 5.02	48.21 ± 4.51	−4.90 ± 0.12	−34.30 ± 4.74
ZINC585291674	−53.01 ± 2.98	−5.26 ± 3.75	23.72 ± 3.58	−3.79 ± 2.75	−38.33 ± 2.95
ZINC585291474	−52.10 ± 3.54	−14.85 ±4.76	34.63 ± 6.13	−3.92 ± 0.14	−36.24 ± 3.69
ZINC257310160	−43.61 ± 4.06	−95.53 ± 12.22	108.53 ± 18.52	−3.67 ± 0.13	−34.28 ± 5.41
ZINC257203083	−41.51 ± 3.84	−88.11 ±11.45	101.01 ± 12.82	−3.76 ± 0.22	−32.38 ± 3.77
ZINC73096242	−54.38 ± 3.20	−28.07 ± 5.31	52.43 ± 5.01	−3.97 ± 0.15	−33.99 ± 3.66

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
