# Peer review of "A Search for Cyclin-Dependent Kinase 4/6 Inhibitors by Pharmacophore-Based Virtual Screening, Molecular Docking, and Molecular Dynamic Simulations"

_ijms, 2021, doi:10.3390/ijms222413423_

Round 1

Reviewer 1 Report

This manuscript describes a computational search for possible new inhibtors of two cyclins, and the analysis of their binding stabilities with molecular dynamics. Several key data regarding the search methodology are, however, missing and (most crucially) no replicates of the MD simulations were performed. Ascertaining the reproducibility of the results is therefore impossible.

Major issues:

Only a single simulation of each compound was performed. This is quite insufficient, since the reader has no way to evaluate the representativeness of the data shown. Every simulation MUST be run at least in duplicate, to confirm (for example) that the influence of  key H-bonds is not an artifact of the initial random assignment of atom velocities.

lines 99-ff.   No data regarding the construction of the ROC curves of the docking protocol was given: how many "active" molecules were present? How did authors ascertain they were indeed "active"? How many decoys were used? How were the decoys generated? Without those data, the reader has now way to confirm that the  ROC curves make sense.

in line 121-122 a word seems to be missing in "These heterocyclic rings and amino chains are a possible site for adenine of  ATP in the hinge region" 

lines 138-139 "According to the docking results, all the hit compounds were successfully docked into the ATP-binding site of CDK4 and CDK6". I cannot tell if this is relevant or not, because the authors do not state  whether or not  the docking box included any other protein regions apart from the ATP-binding site.

The whole pages 6 and 7 are hard to read because the authos do not provide a picture of the active site for the reader to put the information into perspective.

Figures 4-11 are basically unreadable due to the superposition of lines, and devoid of actionable information: global RMSD values are extremely coarse measures which do not allow any reader to understand what is happening throughout the simulations. Authors should, instead , provide graphs showing how the environment around the ligand is evolving : is the active site closing on the ligand, or does it open? are key distances between H-bond donors in the ligand and H-bond acceptors in the protein (or vice-versa) stable or not? etc., etc. etc. 

In pages 11-12, authors show RMSF data and discuss them relative to the ligand binding. This portion of the paper is rendered quite opaque by the lack of any previous description of the protein structure and active site: authors now speak about "hinge regions" and "activation segments" , but the reader has no way to know what they mean because neither of these has been defined previously, nor was the active site placement relative to these regions discussed before.  In lines 229-231, authors state "[These regions are]  quite stable with  low RMSF values. This is probably due to the interaction between the eight hit compounds within these areas, making the compounds have seemingly less fluctuation." This conclusion is not solid: this would only be true if those regions, in the absence of ligands, had noticeably larger RMSFs, and the authors did not show that.

Hydrogen bond analysis should be presented graphically (as graphs of H-bond distance evolution with time). The present description of these data is needlessly opaque.

Section 3.1: In the description of the active molecules used for the pharmacophore analysis, authors state "This training set comprised 13 known active compounds selected from published literature including Palbociclib,  ribociclib, and abemaciclib".  Hard data is missing here: how similar to each other (or not) are the selected active compounds ? How many active compounds are known, and how representative of this set are the selected 13 compounds? Can the selected pharmacophore model correctly identify the active compounds that were not included in the training set?   Line 62 in Section 2.1 also seems to contradict this, since it states that  "Pharmacophore models were built using MOE2014, based on three FDA-approved CDK4/6 inhibitors: palbociclib, ribociclib, and abemaciclib", rather than on "13 known active compounds" (although the "13 known compounds" are mentioned as being part of the model generation in line 68

Section 3.2 : No information regarding the docking box was presented: what were its dimensions? Which regions of the protein were encompassed by the docking box?

Reviewer 2 Report

In this manuscript, the authors have given an account of a ligand-based pharmacophore screening approach of the ZINC15 database followed by docking and molecular dynamics simulations leading to the identification of 2 potential CDK4 and CDK6 binders. Although the hypothesis of targeting CDK4/CDK6 for the treatment of ER-positive breast cancer is reasonable, a number of drawbacks have dimmed the enthusiasm of the reviewer at this stage.

  1. The authors mention that the initial stage towards developing their computational screening method involved screening on the ZINC15 database using a previously generated and validated ligand-based pharmacophore model. However, there is no reference cited for this work.
  2. The authors applied a 2-stage molecular docking strategy using MOE2014 and Autodock4.2. However, it is not clear why this strategy was employed or how using 2 docking softwares vs one would improve the hit rate/quality. There are many reports of identifying hits using either of the above softwares.
  3. Table 2 shows the scores of the hits identified. The two scores do not necessarily rank the 8 compounds in the same order. There is no explanation provided for this observation.
  4. Abbreviations used in Table 1 should be spelled at the end of the table for better readability.
  5. Structures of the FDA-approved drugs should be shown in Figure 3 for comparison to the hits obtained.
  6. There are major mistakes in the summarization of results in the manuscript. For e.g.,
    • Table 2: There are 7 compounds (not six) that show H-bond interaction with Val96
    • The authors mention ‘Of the eight-hit compounds, ZINC585292614 and ZINC257310160 did not form hydrogen bonds with CDK4.’ However, as per Table 2, these compounds form H-bonds with Asp158 and Asp99 respectively.
    • As per Table 3, ZINC585291474 does not form H-bond interactions with Lys43. However, the authors have mentioned this in their discussion.
    • ZINC585292614 also formed an additional hydrogen bond with Lys14, a part of the conserved HRD sequence of kinase domain VIB included in the catalytic loop. - This information is either missing in Table 3 or the Lys residue is misrepresented in the discussion.
  1. Lastly, in the molecular dynamics’ studies section, figures 4, 6, 8, 10, 12, 13 are difficult to read because of the colors used. They should be re-done for more clarity.

Two potential hits were identified by the authors following their computational studies. However, there is no biological (binding) data reported for these. Although the authors mention that these results are on their way, it is critical that the preliminary binding studies be reported in this manuscript to establish the validity of their computational screening method. Currently, this manuscript is only incremental work from the authors’ previously developed ligand-based pharmacophore. Hence it is not recommended for publication in the International Journal of Molecular Sciences.

Round 2

Reviewer 1 Report

While  I appreciate the inclusion of the RMSF graph of the ligand-free protein, authors have generally neither improved the presentation of their results nor included enough detail on the validation of the pharmacophore model:

A) Completely uninformative RMSD graphs are still shown., and the required graphs detailing the evolution of key ligand-active site distanes are still missing. Authors have not included any graphs showing the results of the duplicate runs.

B) the ROC curves clearly show some stepwise increments (in the y axis) much smaller than 0.05.  This is only possible if the number of "active" molecules in their set is much larger than 1/0.05, i.e. much larger than 20. Authors, however, confirm that their data set contains only 13 actives. How can this be?.. With 13 actives, the ROC curve should increase in steps of at least 1/13, or around 0.08. Please explain.

C) Authors seem to confirm that their training set contained 13 actives and 650 decoys. It thus appears that there no compounds were left out of the training to build a validation set. This is not correct: without an independent validation set (i.e. a set whose members were all left out of the training step) all results are self-referential and there is no way if the model works outside the training set.

D) ligand RMSD can refer to two different displacements:

1) the displacement observed upon superposition of ligand in a snapshot over the conformation of the ligand on the reference frame (which only tells us whether the ligand keeps the same conformation but DOES NOT incllude information on its spatial coordinates and is therefore unable to state whether the ligand is moving away ffrom the active site)

2) the displacementof the ligand observed upon superposition of the whole complex in a snapshot with the structure of the whole complex in the reference snapshot. This is the only RMSD that can tell us whether the ligand indeed keeps at approximately the same position. Most automated scripts to compute ligand RMSD DO NOT compute this metric, but only the previous one. Authors should confirm thyat the correct ligand RMSD is being measured and reported on. 

E) Authors fail to show a figure of CDK4/6 wherre the different regions are highlighted. This makes it quite hard to understand their description of these regions in lines 236-251

Reviewer 2 Report

Although the reviewer thinks that preliminary experimental evidence to validate the computational work is necessary, they understand that given the current pandemic it can be restrictive to carry out experimental work. However, while the authors have addressed many of the reviewer’s comments, there are still some critical issues that need to be fixed before publishing to IJMS.

  1. The authors mention: “A total of 150 models containing three to seven combinations of pharmacophore features were obtained.” Detailed information of the combinations along with the validation scores for each model should be summarized as a table in the SI.
  2. It is misleading to add a reference for the same above-mentioned sentence since it could make the reader assume that the work was done earlier. This reference should be removed. (Ref 19)
  3. As suggested previously, figures 4-11 are difficult to read because of the colors used. They should be re-done for more clarity.
  4. Lastly, this manuscript will highly benefit if the authors incorporate a workflow figure in the beginning.

Round 3

Reviewer 1 Report

I am very concerned to see that no validation set was present for the pharmacophore modeling. Authors' claim "We did not carry out
independent validation with other compounds outside the training set because of the limited availability of the active compounds of CDK4/CDK6 inhibitor." is , in this respect, not responsive: authors could have reserved 3-4 molecules (from their initial set of 13) for validation, and built the model from the remaining 9-10.

Authors still show irrelevant backbone RMSD graphs in the main text: these graphs simply demonstrate that the structure remains broadly stable with or withour ligand. This information would would only be useful if there were any suspicion of ligand binding causing dramatic structural changes. Nothing else can be learned from them, and filling two pages of the main text and 5 pages of the supplementary informationwith them is a waste of space. The relevant graphs (which are still missing from the main text) are the evolutions of key ligand-protein distances throughout the simulations, and these are still missing from BOTH the main text and the SI. ooking at the complex structures, selecting 4/5 representative distances beetween ligand atoms and active site features, and plotting them in excel takes (depending on the precise program/script used) at most 30 minutes (and most likely less than 10 minutes) for each simulation. In spite of the ease of producing those graphs (which would be informative, rather than the meaningless backbone RMSD graphs), only two graphs (of H-bond distances) are shown , in the SI, and from a single MD replicate. This is deeply disappointing.

The H-bond distances graphs in the SI are quite strange: the shortest distance they show is 3 angstrom, which is quite a bit larger than common H-bonds. I assume the distance was not measured between H and the accepting electronegative atom, bu rather between the donor electronegative atom and the accepting eelctronegative atom, but this should be stated explicitly. Regardless, those graphs disagree with the data in table 4:  ZINC585292724 is said to have a H-bond to Val96 (CDK4) in 52 % of the time, and the simulation shows that the H-bind distance remains at the minimum vallue for >80 % of the simulation. For ZINC585292587, tabel 4 states 51 % occupancy (i.e. almost the same as ZINC585292724 ) but the graph appears to show much larger variance, etc. etc..... 

Reviewer 2 Report

The authors have now addressed most of the reviewer’s comments. There are a few revisions that still need to be implemented:

  1. The legend of figure 1 can be changed to “Schematic summarizing the overall workflow described in this work.”

‘Inhibitors’ is misspelled in the last text box in this figure. The text box size needs to be made larger for some since it is difficult to read the words in them.

The text box labeled “Pharmacophore model” and “Ligand based pharmacophore modeling” mean the same. In fact, the Pharmacophore model box does not need to be there and should be removed.

  1. As suggested previously and during the first review, figures 5-11 are difficult to read and need to be redone. The authors have only redone one of the 2 plots in each figure. Again, these need to be redone (like ones that they have changed).
  2. Figure 16.: The legends for both the plots are missing units (ns).
  3. In the conclusion section (line 1) the word including should be changed to employing.

Round 4

Reviewer 1 Report

Autors have now provided adequate replies to my queries

Reviewer 2 Report

The authors have addressed all major revisions of the reviewer and the manuscript can now be accepted in the current form.